# Auto-reconfiguration for Latency Minimization in CPU-based DNN Serving

**Ankit Bhardwaj** [1 2]   **Amar Phanishayee** [1 3]   **Deepak Narayanan** [1 4]   **Ryan Stutsman** [5]

## Abstract

In this paper, we investigate how to push the performance limits of serving Deep Neural Network (DNN) models on CPU-based servers. Specifically, we observe that while intra-operator parallelism across multiple threads effectively reduces inference latency, it provides diminishing returns. Our primary insight is that instead of running a single instance of a model with all available threads on a server, running multiple instances, each with smaller batch sizes and fewer threads for intra-op parallelism, can provide lower inference latency. However, the right configuration is difficult to determine manually since it is workload-dependent (DNN model and batch size used by the serving system) and deployment-dependent (number of CPU cores on a server). We present Packrat, a new serving system for online inference that given a model and batch size ($B$) algorithmically picks the optimal number of instances ($i$), the number of threads each should be allocated ($t$), and the batch sizes each should operate on ($b$) that minimizes latency. Packrat is built as an extension to TorchServe and supports online reconfigurations to avoid serving downtime. Averaged across a range of batch sizes, Packrat improves inference latency by $1.43\times$ to $1.83\times$ on a range of commonly used DNNs.

## 1 Introduction

DNN serving is an increasingly important datacenter workload. DNN serving systems are often used in online services like image and video analytics, speech transcription, text and code completion, chatbots, and more. In these settings, requests arrive continuously and must be served in real time; thus, serving systems must handle high request rates efficiently and with low response latency.

There are many DNN serving systems available today, including TensorFlow Serving (Olston et al., 2017), TorchServe (TorchServe Team, 2019), and Triton (NVIDIA Corporation, 2018). These systems are designed to use both CPUs and GPUs to execute DNN model inference. GPUs generally provide better throughput than CPUs, but they are often more expensive and power-hungry. They also end up underutilized for inference workloads (Kosaian et al., 2021). Recent CPU advances, like high core counts (56 to 64 cores are common today (Intel, 2019; AMD, 2022)) and specialized instructions that support lower numerical precision multiplications with higher precision accumulates (AVX-512 (Intel, 2017), AMX (Intel, 2022)), improve inference performance. Every cloud server comes equipped with such multicore CPUs and many product groups at large companies already own large fleets of such servers that are also used for CPU-based serving, including at Meta and Microsoft (Hazelwood et al., 2018; Soifer et al., 2019).

In this paper, assuming the use of CPU-based serving, we push the performance limits of serving DNN models on a single multicore CPU-based server. Serving systems like Triton and TorchServe provide useful features like request handling, adaptive batching of inference requests, and multi-model serving. One important technique to improve the latency of DNN model serving is intra-operator parallelism (Shoeybi et al., 2019), where a single operator is split and run using multiple threads. However, we observe that increasing the number of threads results in diminishing returns; this observation is consistent across batch sizes and DNN models (§2.1). To sidestep the diminishing benefits from intra-op parallelism, a user might try to create one model instance per core and configure their workload to split each batch across the available threads, but as we will show quantitatively later, this does not minimize latency either. In short, neither maximizing intra-op parallelism nor maximizing parallelism across model instances results in the best inference latency.

In this paper, we present Packrat, an optimized CPU-based serving system for online inference that automatically determines the number of threads that need to be allocated to model instances to minimize inference latency. Packrat is motivated by the following key insight: *instead of running a single instance of a model with all available threads (the default for systems like TorchServe), running multiple in-*

---

[1]Work done while authors were at Microsoft Research [2]MIT CSAIL [3]Meta [4]NVIDIA [5]University of Utah. Correspondence to: Ankit Bhardwaj <ankitbwj@mit.edu>.

*Proceedings of the 42^{nd} International Conference on Machine Learning*, Vancouver, Canada. PMLR 267, 2025. Copyright 2025 by the author(s).

*stances each with smaller batch sizes and fewer threads for intra-op parallelism can provide lower inference latency.*

In the general case, determining the optimal configuration of ⟨instances, threads, batch⟩ (or ⟨$i, t, b$⟩ for short) is challenging because it is workload- and deployment-specific. The optimal configuration depends on the specific model being served, input dimensions like the batch size (which is itself dependent on the request arrival rate), and the hardware (e.g., number of cores, memory bandwidth, etc.). Furthermore, even if there were a hypothetical oracle that could provide the optimal ⟨$i, t, b$⟩ configuration, a user would still have to manually recognize when to change configurations and then reconfigure existing serving systems while specifying thread-core affinities appropriately.

Packrat uses a novel algorithm to dynamically determine the optimal ⟨$i, t, b$⟩ configuration for models on individual servers given a batch of inputs for the model. It does this automatically using a small amount of targeted profiling; from this limited profiling information, it formulates ⟨$i, t, b$⟩ configurations that are expected to optimize average batch latency for different batch sizes by solving a 2-dimensional knapsack problem using dynamic programming. This lets Packrat quickly find configurations that balance intra-op latency with multi-instance execution without the need for user input and without impractically profiling *all possible* configuration combinations. Combined with its mechanism of transitioning between configurations, this lets Packrat dynamically reconfigure model instances and threads used for inference, entirely online, so as to optimize inference latency as workloads change. We evaluate Packrat on a single server running TorchServe. Over several models, we show that Packrat improves inference latency and throughput over the baseline approach that maximizes intra-op parallelism by $1.43\times$ to $1.83\times$ averaged over a range of batch sizes. Packrat code is open-source and can be accessed at https://github.com/msr-fiddle/packrat.

## 2 Background and Motivation

DNN inference involves executing a single forward pass of the model for each inference input. The forward pass consists of a sequence of operations like matrix multiplications, convolutions, vector operations, and activation functions that are executed in a specific order. Each request incurs some overhead including data transformations, memory allocations, and data copying. These overheads can be amortized by batching multiple inference inputs and executing them in one forward pass, improving arithmetic intensity and overall performance of the system. To minimize queuing latency, production DNN serving systems like Triton and TorchServe also implement adaptive batching, where if a full batch has not arrived within a timeout, inference execution begins on the requests accumulated so far.

Modern server-class CPUs have 10s to 100s of cores (Intel, 2020) and support multicore parallelism to accelerate inference requests. They also utilize hardware-level optimizations like vector instructions such as AVX-512 and Fused Multiply-Add instructions (Quinnell et al., 2007), which enable efficient execution of matrix operations by processing multiple data elements in parallel. Optimized libraries such as Intel MKL (Wang et al., 2014) and OpenBLAS (Xianyi & Kroeker, 2017) leverage these capabilities to speed up DNN computations, while DNN frameworks like PyTorch (Paszke et al., 2019) and TensorFlow (Abadi et al., 2016) integrate these optimizations to improve inference performance.

### 2.1 Intra-Op Parallelism

As described earlier, each inference involves executing a sequence of operations like matrix multiplication, convolutions, or activation functions with vector operations in a specific order. Each operation can be broken up and executed in parallel across multiple cores. This is called *intra-op parallelism* because operators for a single input's inference (or a batch of them) are executed in a parallel fashion. Depending on the implementation, intra-op parallelism is realized through OpenMP (Chandra et al., 2001) or using MKL threads. By default, OpenMP matches the number of threads it uses to the number of physical cores available on the machine when executing parallel code. However, DNN frameworks like PyTorch and TensorFlow also allow the user to specify the number of threads to use.

To understand the impact of intra-op parallelism on inference performance, we execute inference for different models while sweeping through different batch sizes and number of threads for four different models: ResNet-50 (He et al., 2016), Inception-v3 (Szegedy et al., 2016), GPT-2 (Radford et al., 2019), and BERT (Devlin et al., 2019). We find that for all these models, intra-op parallelism improves inference throughput and latency, but scaling the number of threads assigned to intra-op parallelism provides diminishing returns. For instance, consider ResNet-50 in Figure 1. For a batch size of 4, increasing the number of threads for intra-op parallelism from 2 to 4 improves latency by $1.85\times$, but from 8 to 16 results in a $1.4\times$ improvement. Similarly, for batch size 32, going from 2 to 4 threads improves latency by $1.9\times$, but going from 8 to 16 improves latency only by $1.4\times$.

In summary, our main observation is that while some intra-op parallelism improves inference throughput and latency across models and batch sizes, scaling such parallelism across more cores provides diminishing gains. The key idea in Packrat builds on this observation; stated crudely, Packrat runs multiple instances concurrently and partitions all available threads across these instances with the goal of minimizing inference latency for a given batch size.

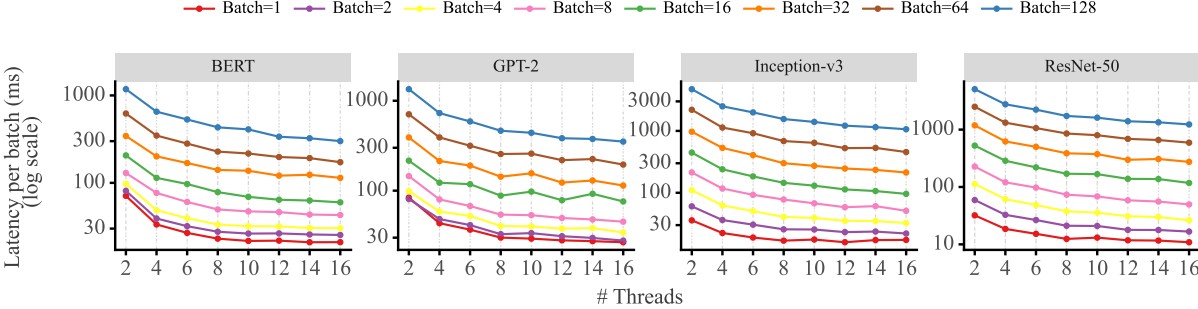

**Figure 1:** Intra-operator parallelism offers diminishing returns after scaling beyond a certain number of threads. Though the exact point and magnitude of diminishing returns may differ, this trend is consistent across different models and batch sizes.

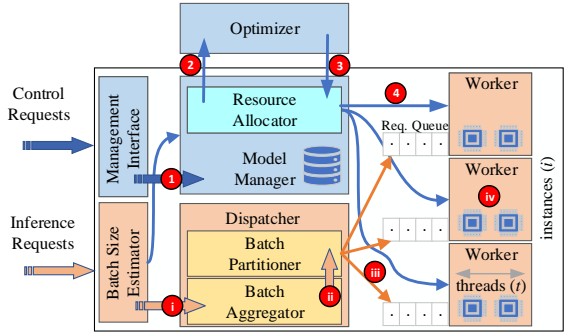

**Figure 2:** Architectural overview of Packrat, highlighting its components and their interactions for the flow of inference requests (orange arrows) and for control messages such as configuration changes (blue arrows).

## 3 Packrat Design

We build Packrat as an extension to TorchServe (a well-established serving system). Packrat is targeted at *online* inference workloads, where input inference requests are constantly streaming in and responses need to be streamed out. When Packrat is enabled in TorchServe, it monitors incoming inference requests to select an appropriate batch size $B$, and transparently and dynamically reconfigures the number of model instances and the intra-op parallelism of each instance to improve average batch latency. In cases where inference request rates change, this configured batch size might need to change as well, triggering reconfiguration.

The key idea in Packrat is that rather than having a single "fat" instance that uses all available threads to parallelize inference within a single batch, it instead divides large batches into smaller batches, each processed concurrently by one of several "thin" instances that use a limited amount of intra-op parallelism. Given a server with $T$ threads and incoming inference requests grouped into batches of size $B$, Packrat determines a configuration $[\langle i_1, t_1, b_1 \rangle, \ldots, \langle i_n, t_n, b_n \rangle]$

such that $\sum_{j=1}^{n} i_j \cdot t_j = T$ and $\sum_{j=1}^{n} i_j \cdot b_j = B$. For simplicity, we will refer to this list as a $\langle i, t, b \rangle$ *configuration* for the remainder of this paper. In each $\langle i_j, t_j, b_j \rangle$ configuration in this list, $i_j$ instances concurrently execute model inference ($i_j$ specifies the number of instances of this type). Each such instance uses $t_j$ threads for intra-op parallelism ($t_j$ is the degree of intra-op parallelism for this instance), and the batch size processed by this instance is $b_j$.

For a given $\langle T, B \rangle$, Packrat tries to pick a configuration that minimizes per-batch latency while improving throughput compared to using $[\langle 1, T, B \rangle]$ (the fat configuration). However, determining the optimal $\langle i, t, b \rangle$ configuration is challenging because it is workload- and deployment-dependent: on the specific model being served, input dimensionality such as tokens in a sequence and batch size, and the number of cores and sockets in the targeted hardware.

To tackle these challenges, Packrat uses a combination of workload profiling, algorithmic techniques to determine the optimal $\langle i, t, b \rangle$ configuration, and systems optimization to seamlessly reconfigure serving instances. Packrat profiles a range of single-instance configurations ($\langle 1, t, b \rangle$ configurations), then uses the measured average batch latency of each single-instance configuration to compute the $\langle i, t, b \rangle$ configuration that minimizes expected average batch latency by using a 2D dynamic-programming-based knapsack solution for a given model and $\langle T, B \rangle$. In some cases, an $\langle i, t, b \rangle$ configuration with only a single element is insufficient to describe the optimal configuration; Packrat handles these cases, but we defer discussion of it to §B for simplicity.

Packrat uses a simple yet effective technique to reconfigure the serving system with an updated $\langle i, t, b \rangle$ configuration without any service downtime (§3.7.1). It maintains two sets of instances, one active and one passive, and it reconfigures the passive set with the desired new configuration. It then swaps the two sets, while scaling up the new active set while simultaneously scaling down old active (now passive) set.

## 3.1 Architecture Overview

Packrat's architecture, in Figure 2, consists of key components that work together to optimize inference latency dynamically. The **Batch Size Estimator (§3.8)** predicts incoming request rates to determine an appropriate batch size. The **Optimizer (§3.3)** then selects the best ⟨i, t, b⟩ configuration using **profiled data (§3.2)**. The **Resource Allocator (§3.4)** assigns compute resources accordingly, while the **Dispatcher (§3.5)** distributes requests among instances. Finally, the **Worker (§3.6)** processes inference requests and returns results. We describe each component in detail next.

## 3.2 Profiling

Packrat uses model profiles to find ⟨i, t, b⟩ configurations that will improve performance for a given ⟨T, B⟩. Model profiling is always done using a single instance at a time, while varying threads for intra-op parallelism (t) and batch size (b). The profiler runs configurations for various ⟨t, b⟩ values. In practice, we use ⟨t, b⟩ ∈ $\{1, \ldots, T\} \times \{2^0, 2^1, \ldots, 2^n\}$. For each of these configurations it records its average batch latency $L_{t,b}$. By using only powers of 2 for b, Packrat reduces the number of profiled configurations from $2^n \cdot T$ to $(n + 1) \cdot T$. Profiling more configurations could lead to more accurate performance estimates (and thus improve the optimizer's final choice of configurations), but §5 shows this modest amount of profiling is sufficient to show substantial gains. Moreover, profiling each configuration takes on the order of minutes, making profiling such a combinatorial space impractical for realistic workloads. For example, for $n = 10$ and $T = 16$, using only powers of 2 for b reduces the number of profiled configurations from 16,384 to 176 which reduces wall-clock profiling time from 30 days to *a few hours*. Profiling is performed offline and not on the inference critical path.

As Figure 3 shows, Packrat's optimizer queries the profiled lookup table to find the expected latency for a given configuration. For each profiled configuration ⟨1, t, b⟩, Profile[t,b] contains the measured single-instance average batch latency (represented as $L_{t,b}$), which the optimizer uses to find configurations to minimize end-to-end latency.

## 3.3 Optimizer

The optimizer is the core component of Packrat. Its goal is to find an ⟨i, t, b⟩ configuration that minimizes average batch latency for a given ⟨T, B⟩. Optimal configurations for a given ⟨T, B⟩ are cached to avoid repeated work.

Packrat uses dynamic programming to find the optimal configuration for a given ⟨T, B⟩, using the latency of the profiled configurations as an input. We use a *multi-dimensional knapsack* problem formulation (Fréville, 2004). The size of the knapsack is 2-dimensional; the first dimension is the

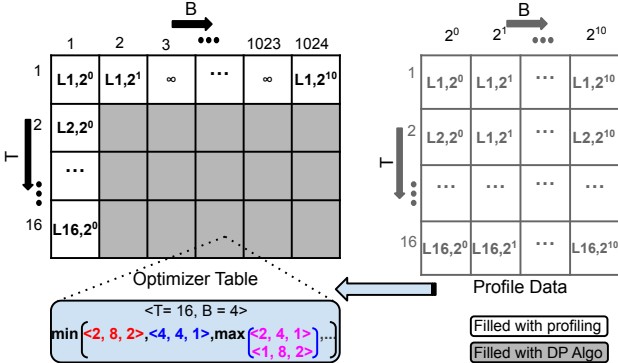

**Figure 3:** Packrat's latency minimization algorithm uses dynamic programming. Single-instance profiled latency data is provided as input to the algorithm, which then fills in a comprehensive optimizer table for all values of ⟨t, b⟩∈⟨T, B⟩.

number of cores (T) and the second dimension is the batch size (B). Profiled configurations are used as the items to fill the knapsack. The weight of each item is ⟨t, b⟩, and the value of the item is the expected average batch latency of the ⟨t, b⟩ configuration. We can use a given ⟨t, b⟩ configuration multiple times (corresponds to the same ⟨t, b⟩ configuration executing concurrently). The goal of the optimizer is to find a set of items that minimizes batch latency (Equation 1) across model instances while keeping the total weight of the items equal to the size of the knapsack ⟨T, B⟩ (Equation 2).

$$\text{Minimize} \max_{\substack{0 \le t_j \le T \\ 0 \le b_j \le B}} L_{t_j, b_j} \qquad (1)$$

$$\text{subject to} \sum t_j \le T \text{ and } \sum b_j = B \qquad (2)$$

$L_{t_j, b_j}$ is the latency of the ⟨$t_j$, $b_j$⟩ configuration. $t_j, b_j$ are the number of cores and batch size of the jth configuration.

We can now describe our dynamic programming algorithm. Let opt[t, b] be the total latency of processing b inputs with the t threads. opt[t, b] has the optimal sub-problem property: we can compute opt[t, b] by looking at opt[t', b'] where $t' \le t$ and $b' \le b$. If possible, opt[t, b] is initialized to the profiled latency with the same number of inputs and threads; otherwise, it is initialized to ∞. Mathematically, opt[t, b] can be computed as follows:

$$\text{opt}[t, b] = \min_{t' \le t, b' \le b} (\max (\text{opt}[t - t', b - b'], L_{t', b'}))$$

where $L_{t', b'}$ is the latency of the profiled configuration ⟨t', b'⟩. The inner max is performed since the end-to-end latency of two concurrent work items is just the latency of the slower work item. The returned configuration is then the one corresponding to opt[T, B]. This algorithm has run-time complexity pseudo-polynomial in T and B, which is practical for reasonable T and B values.

The above algorithm provides the optimal solution in theory since it searches over all possible configurations. However,

in practice, the generated $\langle T, B \rangle$ solution might not match the expected theoretical optimal, since the optimizer depends on profiles measured in isolation, and it disregards performance contention from running various $\langle i, t, b \rangle$ configurations concurrently on the same multicore server (such contention profiling across all configuration combinations is impractical). We show in §A that the gap between the optimal solution in theory and practice is small.

### 3.4 Resource Allocator

The resource allocator assigns resources to instances based on the $\langle i, t, b \rangle$ configuration returned by the optimizer. The resource allocator is the only component that interacts with the dispatcher and the worker. For now, the allocator assumes that resources are not over-subscribed and $\sum i_j \cdot t_j$ is less than or equal to the number of physical cores in the system. Given that the resources are not over-subscribed, the allocator allocates resources to the instances in a round-robin fashion. The compute resources for each instance are statically allocated at the time of instance creation and do not change at runtime. Hence, the allocator pins the instance to the cores allocated to it to avoid thread migration costs.

The allocator is independent of the optimizer and a user can specify other ways to allocate resources to the instances. For example, the user can specify specific cores for each instance. By default, the allocator avoids assigning cores across sockets to prevent performance degradation from inter-socket communication in NUMA domains. However, while individual instances are socket-local, different instances can utilize all available sockets in the system.

### 3.5 Dispatcher

The dispatcher handles two types of requests: (1) management requests and (2) inference requests. The dispatcher's management interface handles "control" messages such as requests to register a new model and those to create and delete instances of any of the registered models. Management requests are handled in the dispatcher itself and are not on the critical path of inference execution.

Inference requests are dispatched to appropriate worker instances. The dispatcher itself handles both batch aggregation and batch partitioning of the requests. Batch aggregation ($B$) is done per model and batch partitioning is done per instance using the $b$ values in the $\langle i, t, b \rangle$ configuration. Batch aggregation also uses a user-provided batch timeout value; request aggregation is done until the timeout expires. If the timeout expires before the batch size $B$ is reached, the dispatcher simply dispatches the current batch to the instances. The Batch Size Estimator triggers a configuration change if timeouts happen frequently. However, instance reconfiguration is time-consuming and is done conservatively.

### 3.6 Worker Instance

Each worker instance is responsible for executing an inference batch with $b$ inputs for a given model using $t$ threads. Each worker executes a user-provided handler over a batch of requests. A handler takes the batch of requests as input and returns the batch of responses. During the handler initialization, the worker might need to load the model into memory. Users may also specify any optimizations to use during model initialization. For example, the user can specify that the model should be loaded and optimized for graph mode (TorchScript for the PyTorch framework).

Inference is executed by the framework using parallel implementations of the operators (intra-op parallelism). Each parallel operator implementation is responsible for executing the operator across $t$ intra-op threads. This parallelization involves slicing the input batch into multiple chunks, partitioning operator state across threads, and executing the operator on each chunk in parallel. Packrat does not improve the mechanism of operator parallelization but simply uses the functionality provided by the framework in a more efficient way by assigning an appropriate number of threads.

### 3.7 Configuration Changes

Reconfiguration is the process of changing the $\langle i, t, b \rangle$ configuration for a model and is handled by Packrat's Resource Allocator. The Batch Size Estimator triggers a configuration change by invoking the optimizer with a new batch size $\widetilde{B}$ if it predicts that the request arrival rate for a given model has changed considerably. Reconfiguration does not require Packrat to run any new profiling; as the batch size changes, the optimizer is re-run with the new $\widetilde{B}$ value to find the right configuration for the new batch size (if the given $\langle T, B \rangle$ configuration is not present in the optimizer cache).

Reconfiguration is time-consuming and done conservatively (§3.8). Packrat works with an implicit assumption that the workload for a given model does not change frequently, which is a reasonable assumption for many datacenter workloads (Hazelwood et al., 2018). Moreover, dramatic workload changes would not only affect Packrat configuration but could also require datacenter-level resource re-provisioning.

Packrat uses a TorchServe feature, worker scaling, to handle configuration changes. Worker scaling is the process of increasing or decreasing the number of workers for a given model. However, in Packrat, we might have to change the configuration of the model instance itself by allocating it fewer or more threads than currently assigned.

Packrat handles the configuration change in two different ways. The first is when a configuration change only requires increase or decrease in the number of instances, but the number of threads within each of the existing instances remains the same. Such configuration changes are handled

by the worker scaling mechanism. Scaling down is achieved by removing the workers of a model one by one. Workers are removed in a round-robin fashion and resources are released back to the resource allocator (§3.4). Scaling up is similar to the initial worker creation process.

The second is the trickier case, and it occurs when the configuration change requires different number of threads for the workers as compared to their current configuration. Packrat handles such reconfigurations in a two-step process called active-passive scaling (§3.7.1). Packrat relies on this process to avoid changing the internal operator implementation libraries and making our approach portable across serving systems. Operator implementation libraries like ATen, MKL-DNN, etc. have their own internal mechanisms to manage and schedule the threads (OpenMP, 2020), but these libraries are not designed to handle frequent configuration changes due to associated higher overheads (Gross, 2018).

### 3.7.1 ACTIVE-PASSIVE SCALING.

Packrat uses active-passive scaling when the optimizer's suggested configuration change requires instances to adjust the number of threads allocated to instances. A naive way of going about such a reconfiguration would be to first shut down all instances in the old configuration (e.g, $\langle i_1, t_1, b_1 \rangle$) and then start all instances in the new configuration (e.g., $\langle i_2, t_2, b_2 \rangle$). In the worst case, all the old workers will be removed, and new workers will be created. However, such an approach risks having the serving system be unresponsive for the entire duration of such reconfigurations.

Packrat uses active-passive scaling to avoid disruption. For each model, Packrat maintains two versions of the model. The active version respects the current configuration and is currently serving requests. The passive version has zero workers and stays inactive until activated.

Active-passive scaling is done in three steps. First, the passive version is scaled up to the new configuration (e.g., we scale up to $i_2$ workers as per the new $\langle i_2, t_2, b_2 \rangle$ configuration). Next, the dispatcher starts redirecting new requests to the new passive instances. Finally, the historically active version is scaled down to zero workers in the background (from $i_1$ workers as per the old $\langle i_1, t_1, b_1 \rangle$ configuration) once they have completed their ongoing requests and been deactivated at the dispatcher. At this point, the active and passive sets of workers have been swapped.

### 3.8 Batch Size Estimation

To choose a good configuration, Packrat needs to know the batch size for the current workload ($B$). Packrat estimates the batch size in an online fashion by tracking the request queue depth over time. It is easy enough for the Batch Aggregator to track the size of each batch that it passes

to workers, but this batch size varies over time depending on input request arrivals, and different batch sizes have different "optimal" $\langle i, t, b \rangle$ configurations.

Reconfiguring the number of instances and threads takes several seconds and is expensive (§5.2.2), so it is important that reconfiguration only happens when the workload is stable enough to warrant it. Without some kind of smoothing, Packrat will risk "flip-flopping" between configurations. Packrat uses two-level smoothing to avoid this problem. First, Packrat's Batch Size Estimator uses the most recent request queue depth $\widehat{Q}$ to track an exponentially weighted moving average of request queue depth ($\widetilde{Q}_x = \alpha \widehat{Q} + (1-\alpha)\widetilde{Q}_{x-1}$) and picks the next lower power of two to $\widetilde{Q}$ as an estimated batch size $\widehat{B}_x$. Second, Batch Size Estimator takes the mode over the last $n$ estimated batch sizes ($\widehat{B}_{x-n}, \ldots, \widehat{B}_x$) to get a final smoothed batch size ($\widetilde{B}$). After each reconfiguration timeout, Packrat's Batch Size Estimator compares the current batch size $B$ to the smoothed batch size $\widetilde{B}$. If $\widetilde{B}$ is different from $B$, Packrat reconfigures the system to use the new batch size $\widetilde{B}$. §5.2.2 shows that this approach works well in practice, and Packrat uses it to both scale up and scale down the batch size $B$ as request arrival rates change.

## 4 Implementation

We implement Packrat as an extension to TorchServe, a popular serving system in the PyTorch ecosystem. Packrat augments TorchServe with features such as the batch size estimator, batch aggregator, optimizer, and resource allocator. TorchServe supports plugins to customize the serving system. For example, we customize the batching layer to provide our custom batch aggregation, batch partitioning, and batch size estimation strategies, and the worker management layer for our custom resource allocator. In all, Packrat is implemented in $\sim$ 5k lines of code.

**Optimizer.** The optimizer is responsible for providing the optimal configuration for a given $\langle T, B \rangle$ pair. However, it does not directly interact with the resource allocator. We implement the optimizer as a standalone service. A separate task acts as a client to the optimizer and uses TorchServe's management API to communicate with the resource allocator. The resource allocator then updates the configuration to match the desired configuration returned by the optimizer.

**Resource allocation.** TorchServe supports custom resource allocators that can create and destroy workers or allocate resources to workers. For fault tolerance, TorchServe also re-spawns workers if they die. Packrat's custom TorchServe resource allocator maintains information about all idle and busy cores and the desired Packrat configuration. Based on the target configuration, it creates workers on demand, and it destroys them when they are not needed. Internally, it uses a modified version of Intel's IPEX TorchServe launcher to

**Table 1:** Server configuration for all our experiments.

| | |
|---|---|
| **CPU** | 2× 16-core Intel Xeon Gold 6142 at 2.6 GHz |
| **RAM** | 384GB (6x32 GB DDR4-2666 DIMMs/Socket) |
| **OS** | Ubuntu 20.04 LTS, Linux 5.4.0-100-generic |
| **Software** | Python 3.8.10, PyTorch 1.12.1, TorchServe 0.6.1, Intel MKL-DNN v2.6.0, OpenMP 4.5 |

launch and pin worker threads on the desired cores (Intel Extension for PyTorch, 2022). This custom resource allocator is only used when Packrat is enabled in TorchServe.

**Batch aggregation and request estimation.** TorchServe allows integrating custom batching algorithms at startup time. Our batch aggregator extends TorchServe's default implementation; we add a batch size estimator that intercepts incoming requests and estimates the batch size for each inference endpoint (§3.8).

## 5 Evaluation

We evaluate Packrat using both inference microbenchmarks with PyTorch and end-to-end performance with TorchServe. Our evaluation seeks to answer the following questions:

- How does Packrat's proposed approach compare to the state-of-the-art for inference microbenchmarks with PyTorch for a range of models and batch sizes? (§5.1.1)
- How does Packrat improve end-to-end serving latency and throughput in TorchServe, and how do these numbers compare to microbenchmarks with PyTorch? (§5.2.1)
- How effective is Packrat's reconfiguration in avoiding stalls and improving serving latency? (§5.2.2)
- How accurate is the optimizer in predicting the performance of multi-instance configurations? (§A)

**Experimental Setup:** All experiments are performed on a single Cloudlab (Duplyakin et al., 2019) c6420 machine (see Table 1) with hyperthreading disabled to avoid interference (Cho & Saroufim, 2022). Benchmarks report time averaged over 100 iterations.

**Models:** We benchmark ResNet-50, Inception-v3, GPT-2, and BERT models. ResNet-50 and Inception-v3 models are image classification models, GPT-2 is a text generation model, and BERT is a text classification model. These models are widely used in real applications.

### 5.1 Microbenchmarks

**Setup.** We microbenchmark various PyTorch models (in eager and graph modes) and batch sizes using pretrained models from the model zoo. First, we report the inference time using PyTorch (with and without Packrat). Later in §5.2, we report the end-to-end latency of inference requests when using TorchServe (with and without Packrat). Unless explicitly stated, we report Packrat speedup numbers compared to corresponding fat-instance baselines. Similarly,

unless explicitly called out, we only report the graph mode results as it consistently provides lower latency than eager mode in all our experiments, and thus is a stronger baseline.

### 5.1.1 SPEEDUP OVER BASELINE EXECUTION.

Figure 4 shows the throughput and latency speedup of multi-instance execution over fat instances for ResNet-50 (a), Inception-v3 (b), GPT-2 (c), and BERT (d). The speedup is measured for different batch sizes and for all threads in a socket. The fat instance is run with 16 threads and batch size $B$ and the thin instances use the $\langle i, t, b \rangle$ configuration suggested by Packrat's optimizer where $\langle T, B \rangle$ is partitioned across $\sum i_j$ smaller instances where $\sum i_j \cdot t_j = T$ and $\sum i_j \cdot b_j = B$. For a given $\langle T, B \rangle$, we measure the average throughput and latency of Packrat's chosen configuration ($\tau_P$ and $\lambda_P$) and of the fat-instance baseline ($\tau_B$ and $\lambda_B$). Throughput and latency speedups are calculated as $\tau_P/\tau_B$ and $\lambda_B/\lambda_P$, respectively. In practice, our throughput and latency speedup are almost always the same.

Even though Packrat's chosen configurations use the same total number of threads as the fat instance, Packrat obtains substantial improvements in latency and throughput. The image classifiers, ResNet50 and Inception-V3 show a 1.53× and 1.52× mean speedup across batch sizes, respectively; the language models GPT-2 and BERT show a 1.18× and 1.13× average speedup, respectively.

**Speedup Reasons:** There are two key reasons that Packrat's configurations outperform the fat instance which uses all threads on the server for intra-op parallelism. First, all OpenMP threads synchronize at multiple barriers in the fat-instance execution resulting in compute resource under-utilization. However, in multi-instance execution, thread(s) in each instance can execute independently of other instances, allowing the multi-instance execution to efficiently utilize the compute resources. Second, usually, workloads have multiple phases with different characteristics (e.g., a part that is compute-intensive and another that is memory-intensive). OpenMP barrier sync enforces all the threads to march in lock-step, forcing every thread to execute similar work. This results in over-utilization of one resource and under-utilization of other resources. However, Packrat's configurations include some degree of multi-instance execution; hence, the threads in each instance can execute different phases without coordination. This results in better average compute and memory bandwidth utilization, which is also apparent when profiling the execution of both approaches.

Packrat's profiler does not exhaustively profile all possible $\langle t, b \rangle$ configurations to keep total profiling time on order of hours (instead of days). We found empirically that exhaustive profiling (all values of $b$) does not change the Packrat-selected configurations and hence actual latency speedups.

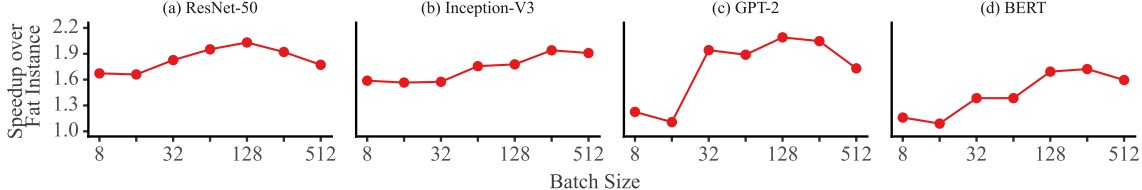

**Figure 4:** Inference microbenchmark when using Packrat in PyTorch, showing the **speedup** of Packrat over baseline fat-instance execution in graph mode for four different DNN models. We also see the comparison of Packrat's expected speedup (estimated from isolated runs of individual instances) to the actual speedup attained when running multiple thin instances concurrently.

**Figure 5:** Packrat's end-to-end latency and throughput speedup over the corresponding baseline (fat-instance) runs in TorchServe. GPT-2 is the only model that uses eager mode in this figure; it crashes TorchServe in graph mode.

**Comparison with single-threaded instances:** Another easy-to-configure baseline is to run $T$ single-threaded instances, one on each CPU core. Single-threaded instances perform worse than the fat instance baseline because of increased cross-instance resource interference and under-utilization (§A). Moreover, single-threaded instances also inherently cannot exploit intra-op parallelism to improve latency. In contrast, Packrat always finds high-performing configurations and provides some speedup against $T$ single-threaded instances (figure omitted due to space constraints). For GPT-2 Packrat performs up to $3.2\times$ better and for other models, Packrat performs either better (from $1.02\times$ to $1.75\times$) or equal to 16 single-threaded instances.

The difference between expected and actual speedups arises from increased resource contention due to multiple instances running simultaneously. In Appendix A, we provide a detailed explanation of the underlying causes. Further exploration in Appendix C demonstrates that this contention is not as pronounced with newer AMD and Intel server machines.

### 5.2 End-to-End Experiments

Next, we evaluate Packrat's end-to-end performance on TorchServe using the same setup as in §5.1. Unless stated otherwise, we use the default TorchServe and report results for the graph mode due to its better baseline performance.

#### 5.2.1 LATENCY IMPROVEMENTS.

Figure 5 shows the latency and throughput speedup of Packrat's configurations over baseline fat-instance execution for ResNet-50 (a), Inception-v3 (b), GPT-2 (c), and BERT (d). Packrat consistently improves performance across all batch sizes for all models. Packrat provides an average speedup

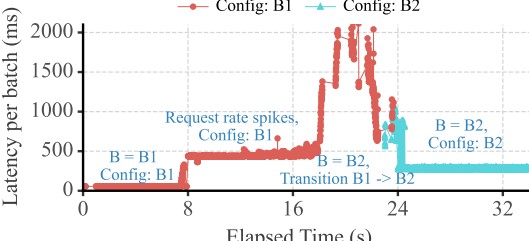

**Figure 6:** Configuration change in Packrat. A request rate spike at $8\,\text{s}$ increases latency under configuration $B1$. After reconfiguration to $B2$ at $24\,\text{s}$, Packrat stabilizes and improves serving latency.

of $1.43$ to $1.83\times$ and a maximum speedup of $1.72$ to $2.09\times$.

**Gains compared to microbenchmarks.** Packrat's end-to-end gains on TorchServe are higher than in the microbenchmarks. This is because the microbenchmarks only perform inference; with Packrat on TorchServe, we measure end-to-end impact on both inference and TorchServe's serving components as well. For each inference batch, a worker executes a handler that consists of pre-processing, inference, and post-processing. Since the pre- and post-processing are usually not compute intensive, serving systems typically use a single thread for them. However, multi-instance execution parallelizes pre- and post-processing as well, resulting in higher performance gains compared to microbenchmarks.

#### 5.2.2 CONFIGURATION CHANGE LATENCY.

Finally, we evaluate the impact of configuration change using the Inception-v3 model, with input arrival rates following a step function. All of the cores on a single CPU socket are used for this experiment ($T = 16$). Figure 6 shows a zoomed-in timeline of model latency just before, during, and just after a configuration change. The experiment starts with the multi-instance configuration for $B = 8$, which correctly corresponds to the load generated by the

client and produces the expected batch size. After some time, the request rate spikes, changing the ideal batch size for the workload to $B = 64$; however, we force the server to not activate a change in batch size immediately (to observe the performance impact of doing this) and the server continues to handle these batches with the $B = 8$ optimized configuration for some time. Finally, the server starts the configuration change for $B = 64$ and begins handling requests with the new configuration.

There are five key takeaways in Figure 6: (1) Response latencies are initially stable. (2) After $8\,$s, the client increases the input request rate, and the server handles requests with the old configuration until $18\,$s. The average latency increases significantly due to queuing delays. (3) After $18\,$s, the server starts the configuration change for $B = 64$. The average latency per batch increases due to the configuration change overhead until $23\,$s; the serving system does not stall processing requests during reconfiguration. The configuration change takes around $5\,$s. Most of this comes from underlying systems (confirmed with a microbenchmark on TorchServe). Given that such reconfigurations occur infrequently, we consider this overhead reasonable. (4) During reconfiguration ($18\,$s-$23\,$s), the server handles requests with the new configuration. The average latency per batch jumps by 2-3$\times$ due to initialization overhead of the new configuration and resource oversubscription, as both old and new configurations are active. (5) After initialization completes, the server handles all requests with the new configuration, reducing latency by $1.54\times$ over the old configuration.

## 6  Related Work

In this section, we describe related work relevant to Packrat.

**Adaptive batching.** Many recent works optimize batched network request processing (Belay et al., 2014; Kaffes et al., 2019; Prekas et al., 2017; Ousterhout et al., 2019) or packet processing (Lévai et al., 2020; Bhardwaj et al., 2017) on multi-core CPUs. Similar to Packrat, these systems process up to $B$ requests to completion before collecting the next batch. These systems face a similar problem in how they divide work across cores; for example, MICA (Lim et al., 2014) showed that some workloads benefit from strict per-core state and work partitioning, while others favor balancing request processing across cores and letting them all synchronize access to shared state. Packrat's techniques may be useful for navigating that trade-off automatically.

Adaptive batching has also been used to serve DNN models more efficiently. Clipper uses adaptive batching to maximize throughput subject to a compute processing latency (Crankshaw et al., 2017). Nexus tries to schedule the inference computations of multiple models onto a given set of resources while respecting a provided throughput and la-

tency SLO (Shen et al., 2019). It determines the right batch size for each model while obeying latency and throughput constraints and simultaneously minimizing the number of resources used. InferLine uses adaptive batching to determine how best to satisfy latency SLOs for pipelines consisting of one or more ML models (Crankshaw et al., 2020), and others have used adaptive batching to support inference with serverless systems (Ali et al., 2020). Triton and TorchServe support adaptive batching and concurrent model serving on the same GPU for better memory and compute utilization.

**Configuration optimizations.** McBench (Wang et al., 2020) uses internal model knowledge and inputs to generate TensorFlow configurations. TensorTuner (Hasabnis, 2018) applies gradient-based optimization to TensorFlow's threading; however, it does not partition an input batch across instances, support reconfiguration, or target optimizing inference latency in the context of serving systems. TVM (Chen et al., 2018) and Tensor Comprehension (Vasilache et al., 2018) are compilers that generate optimized code for operators on various hardware backends. ParaX (Yin et al., 2021) advocates for single-threaded inference to avoid stalls.

**CPU-based optimizations.** Many optimizations have helped reduce inference latency of popular models. For example, removing zero padding and using tensors with dynamic shapes, using optimized matrix multiplication libraries, fusing memory-bound operators to increase arithmetic intensity, and effectively using vector units (Dice & Kogan, 2021; Le, 2020; Lv, 2019; Daghaghi et al., 2021; Liu et al., 2019). Many such techniques are part of optimized libraries such as Intel's MKL that are used in PyTorch, and they help bolster the baseline that Packrat improves over.

**Production serving systems.** Amazon SageMaker, Azure ML, and TensorFlow Serving (Services, 2020; Microsoft Azure, 2022; Olston et al., 2017) simplify model deployment, including serving various model versions with minimal overhead. However, these systems do not consider how models should be partitioned over a multi-core server, and the effect of $\langle i, t, b \rangle$ configuration on end-to-end latency.

## 7  Conclusion

Minimizing CPU-based inference latency for a given workload is challenging. Pure inter- and intra-op parallelism results in sub-optimal latency. Moreover, the best configuration depends on the model and the CPU hardware. Packrat solves this using an automated approach that combines selective profiling, an optimizer that estimates the performance of unprofiled configurations and suggests configurations to minimize latency, and performs online reconfigurations to avoid serving downtime. Collectively, these let Packrat realize latency and throughput speedups of $1.43\times$ to $1.83\times$ averaged across batch sizes on a range of common DNNs.

## Acknowledgments

Ankit Bhardwaj contributed to this work as a PhD student at the University of Utah and during an internship at Microsoft Research. This material is based upon work supported by the National Science Foundation under Grant No. CNS-2245999. Any opinions, findings, and conclusions or recommendations expressed in this material are those of the authors and do not necessarily reflect the views of the National Science Foundation.

## Impact Statement

This paper presents work whose goal is to advance the field of Machine Learning. There are many potential societal consequences of our work, none of which we feel must be specifically highlighted here.

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

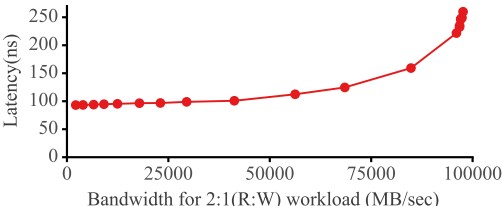

**Figure 7:** Effect of memory bandwidth load on access latency.

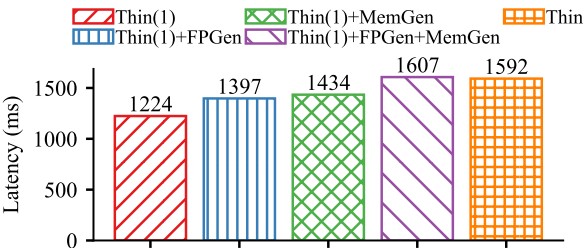

**Figure 8:** Breakdown of why multi-instance performance does not match the performance predicted from profiling single-instance latency (Thin(1)). When combined with a SIMD (Thin(1)+FPGen) or a memory bandwidth load generator (Thin(1)+MemGen) or both (Thin(1)+FPGen+MemGen) a single instance's performances slows to match that measured when 16 instances run together (Thin).

## A Expected versus Observed Speedup.

Figure 4 shows that while Packrat is able to obtain significant latency benefits over the baseline, the attained speedup for all models (across batch sizes) is less than Packrat's expected speedup – this can be seen in the gap between the Expected Speedup and Actual Speedup lines. This is because Packrat estimates the performance of configurations composed of multiple instances running concurrently based on profiling of individual instances run in isolation; however, the actual performance of each instance when run concurrently with other instances is lower since concurrent execution creates some resource contention between the instances. For example, running Packrat's configuration in practice creates more CPU and memory bandwidth interference compared to when profiling was done. Next, we dive deeper to understand and account for these sources of contention.

**License-based downclocking.** The first major source of interference between instances is due to license-based CPU downclocking (Schöne et al., 2019). License-based downclocking is a mechanism CPU vendors use to limit CPU frequency when many cores use SIMD instructions concurrently for sustained periods. This is done for energy efficiency reasons (Schöne et al., 2019). For example, even though the normal CPU frequency for an Intel Xeon Gold 6142 is 2.6 GHz (Table 1), its frequency is downclocked to 2.2 GHz when all cores run SIMD instructions concurrently (WikiChip, 2017). This lowers each core's performance by about 15%; we experimentally show this explains about half of the gap between expected and observed speedups below.

**Loaded memory latency.** The second source of interference is due to increased load on the memory controller. Ideally, cores would be able to use any available memory bandwidth without impacting other cores so long as memory bandwidth isn't saturated; however, in practice, memory bandwidth load created by one instance increases effective memory access latency for other instances. Figure 7 shows this effect by measuring memory access latency under varying memory bandwidth load; this microbenchmark uses a 2:1 read-write ratio similar to our inference workloads. The increased memory latency explains the other half of the gap between expected and observed speedups.

**In-depth analysis for ResNet-50 microbenchmark.** To verify that downclocking and degraded memory access latency explain the difference between expected and actual performance, we perform an in-depth analysis using the ResNet-50 microbenchmark. As shown in Figure 4 (a), due to the overheads of the multi-instance execution the gap between expected and realized speedups is between 12-15%. To show the impact of license-based downclocking on inference performance, we implement a SIMD load generator that saturates the FMA units on a configurable number of cores. We monitor the performance of a single thin instance while we run the SIMD load generator on the cores that are not being used for inference (Figure 8, Thin(1) + FPGen).

Similarly, to show the effect of increased memory latency on inference performance, we implement a custom load generator that generates a configurable amount of memory bandwidth load. We use this to generate load that is about equal to the load generated by $i - 1$ thin instances, which simulates the memory load of running a thin instance concurrently with $i - 1$ other instances (Figure 8, Thin(1) + MemGen).

Figure 8 shows this analysis for a single configuration ($T = 16$, $B = 256$). The optimizer recommends using 16 thin instances each with ($t = 1$, $b = 16$). The latency for the baseline fat instance is $2664\,\mathrm{ms}$ and for a single ($t = 1$, $b = 16$) instance is $1224\,\mathrm{ms}$ which is about a 54% reduction over the baseline. However, when we use 16 thin instances, the latency is $1600\,\mathrm{ms}$ which is about a 40% reduction over the baseline fat instance. So, the actual latency reduction is 14% lower than the expected reduction. The goal of this analysis is to understand the gap between single thin instance (Thin(1)) and multiple thin instances (Thin) as shown in Figure 8. The impact of license-based downclocking increases the latency of a single thin instance to $1397\,\mathrm{ms}$ (Thin(1) + FPGen) and the impact of increased memory latency degrades the latency of Thin(1) to $1434\,\mathrm{ms}$ (Thin(1) + MemGen); these

**Table 2:** The best $\langle i, t, b \rangle$ configurations identified by Packrat's Optimizer for BERT with different batch sizes for two deployments with $T = 16$ and $T = 14$ respectively.

| Batch Size ($B$) | Cores ($T = 16$) | Cores ($T = 14$) |
|---|---|---|
| 8 | $\langle 2, 8, 4 \rangle$ | $\langle 2, 7, 4 \rangle$ |
| 16 | $\langle 4, 4, 4 \rangle$ | $\langle 1, 6, 8 \rangle, \langle 2, 4, 4 \rangle$ |
| 32 | $\langle 4, 4, 8 \rangle$ | $\langle 1, 6, 16 \rangle, \langle 2, 4, 8 \rangle$ |
| 64 | $\langle 4, 4, 16 \rangle$ | $\langle 1, 6, 32 \rangle, \langle 2, 4, 16 \rangle$ |
| 128 | $\langle 4, 4, 32 \rangle$ | $\langle 2, 7, 64 \rangle$ |
| 256 | $\langle 8, 2, 32 \rangle$ | $\langle 2, 3, 64 \rangle, \langle 4, 2, 32 \rangle$ |
| 512 | $\langle 8, 2, 64 \rangle$ | $\langle 2, 3, 128 \rangle, \langle 4, 2, 64 \rangle$ |
| 1024 | $\langle 8, 2, 128 \rangle$ | $\langle 2, 3, 256 \rangle, \langle 4, 2, 128 \rangle$ |

are 173 ms and 210 ms higher (worse) than the isolated single thin-instance latency. As reference points in Figure 7, Thin(1) generates memory traffic of around 3 GB/s and Thin generates around 50 GB/s. If we add all three overheads, we get around 1600 ms, which is the latency of multiple thin instances latency (Thin). Hence, the combination of license-based downclocking and increased memory access latency explains the discrepancy between the expected latency estimated from profiling thin instances in isolation and the actual latency in practice.

**Why not model resource interference in the optimizer?**
Packrat's Optimizer does not model the interference described in §A. While designing Packrat, we hypothesized that since this interference affects all configurations in a similar way, modeling interference wouldn't change the selected configurations.

To validate this hypothesis, we did two things. For select models, we looked at the gap between actual and expected speedup (e.g., in Figure 4). In all cases, actual is a constant factor slower than expected (reassuringly, the *same* constant factor across models). Next, for these models, we reran the Optimizer's DP taking into account the estimated interference performance penalty. The resulting configurations (and actual performance) were identical to the configurations selected when not accounting for interference. The reason is that if all profiled performance measurements are penalized by multiplying them by some constant $c < 1$, then their relative order doesn't change. The equation for opt$[t, b]$ still makes the same choices at each step even if all costs are multiplied by the same $c$. Packrat could be extended to incorporate modeled interference, but we have yet to find a case where doing so changes the chosen configuration.

## B  Non-Uniform Instances in Packrat.

For most of the benchmarks, the optimizer generates a uniform $\langle i, t, b \rangle$ configuration where each thin instance has the same number of cores and batch size. This is because the number of threads ($T$) and batch size ($B$) used in most benchmarks are powers of two. In real-world scenarios, number of cores and batch sizes may not always be pow-

**Table 3:** Additional server hardware configurations to demonstrate Packrat's performance gains beyond the primary testbed.

| | |
|---|---|
| **Server 1** | **CPU**: 16-core AMD EPYC 7302P at 3.0 GHz 
 **RAM**: 128 GB (8×16 GB DDR4-3200) |
| **Server 2** | **CPU**: 28-core Intel Xeon 5512U at 2.1 GHz 
 **RAM**: 128 GB (8×16 GB DDR5-4800) |

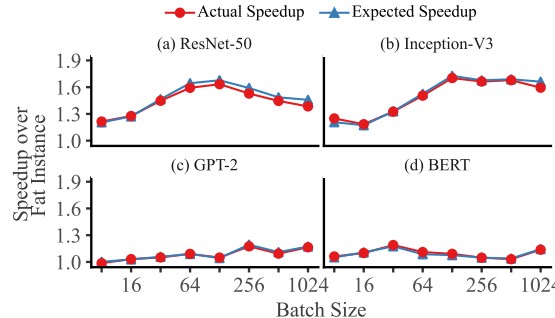

**Figure 9:** Multi-instance performance on an EPYC machine.

ers of two; hence, we investigate the performance in these cases here. We show the impact of such configurations on the inference latency and throughput. To conserve space, here we show the results only for the BERT model. Results for other models are similar. Table 2 shows the $\langle i, t, b \rangle$ configurations for different batch sizes for T = 16 and T = 14. For example, for $B = 16$, the configuration is ($i = 4$, $t = 4$, $b = 4$) for T = 16; it is ($i = 1$, $t = 6$, $b = 8$) and ($i = 2$, $t = 4$, $b = 4$) for T = 14. So, the final configuration includes a mix of different thin instance types. Similar configurations are generated for other batch sizes. For such cases, Packrat's optimizer chooses configurations where the latency of different instances types are similar, resulting in lower overall latency while satisfying Equations 1 and 2.

## C  Evaluating Packrat on Different Machines

Packrat is designed to be a general-purpose system for different machine configurations. To demonstrate its performance across different hardware configurations, we evaluate Packrat on two additional Cloudlab (Duplyakin et al., 2019) machines with different CPU architectures and memory configurations (as shown in Table 3). The first machine is an AMD EPYC 7302P with 16 cores and 128 GB of RAM (Cloudlab c6525-25g), while the second machine is an Intel Xeon 5512U with 28 cores and 128 GB of RAM (Cloudlab c6620). Both machines have 8×16 GB of DDR4-3200 and DDR5-4800 memory, respectively. We run the same set of microbenchmarks as in §5.1 on these machines.

Figures 9 shows the results for the AMD EPYC machine. While Packrat consistently outperforms the baseline, the performance varies in two ways compared to the primary setup. First, the baseline and the Packrat configurations

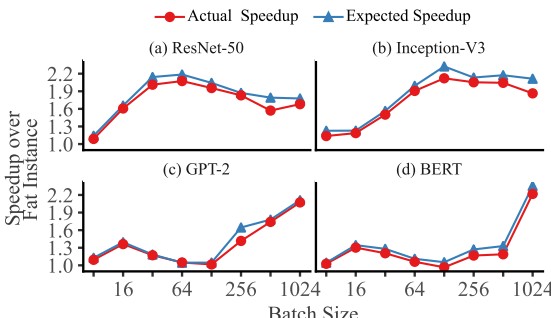

**Figure 10:** Multi-instance performance on an Emerald Rapids machine.

achieve lower performance than the primary setup. Second, the difference between expected and actual vanishes for most models and batch sizes because the memory bandwidth is not a bottleneck on these machines (except for a few large batches), and AMD EPYC does not support AVX512. Thus, the CPU does not experience any downclocking.

Figure 10 shows the results for the Emerald Rapids machine. The absolute performance is higher than the EPYC machine. These machines also have a higher memory bandwidth, and improved licensed-based downclocking support for AVX512 instructions. As a result, the difference between expected and actual is minimal as compared to the primary setup. Overall, Packrat consistently outperforms the baseline on both machines across models and batch sizes.

