# OpenReview forum: "Auto-reconfiguration for Latency Minimization in CPU-based DNN Serving"
_ICML.cc/2025/Conference — ICML 2025 poster_

### Official Review · Reviewer_9PcW · 2025-02-26

**Overall Recommendation:** 3

**Summary:**

This manuscript investigates methods for accelerating neural network model tasks on CPU-based servers to minimize latency. Specifically, the authors found that current frameworks such as TorchServe, although effective in reducing inference latency through intra-operator parallelism across multiple threads, exhibit diminishing returns. Therefore, the authors propose that instead of running a single instance of a model utilizing all available threads on a server, running multiple instances, each with smaller batch sizes and fewer threads for intra-operator parallelism, can provide lower inference latency. They also propose a corresponding algorithm to identify the optimal configuration for minimizing latency. Finally, extensive experimental validation is conducted, demonstrating the effectiveness of the proposed framework. The topic is of interest and the presented numerical results seem, indeed, promising.

**Claims And Evidence:**

Yes

**Essential References Not Discussed:**

No

**Experimental Designs Or Analyses:**

The author conducted experiments on a single CloudLab and validated the results using tasks involving ResNet-50, Inception-v3, GPT-2, and BERT models. Specifically, the study first compared throughput and latency acceleration with baseline methods and analyzed the underlying reasons. Additionally, the author provided the latency of configuration changes in the algorithm.

**Methods And Evaluation Criteria:**

Yes

**Other Comments Or Suggestions:**

Please refer to **Questions For Authors**.

**Other Strengths And Weaknesses:**

**Strengths:**
- The authors provide a detailed modeling process.
- The authors conducted extensive experimental validation across a wide range of models.
- Based on the experimental results, it can be concluded that this research holds certain significance for serving Deep Neural Network (DNN) models on CPU-based servers.
- This algorithm is built as an extension to TorchServe and supports online reconfigurations to avoid serving downtime.

**Weaknesses:**
- There is a lack of sufficient theoretical proof.
- Other, please refer to **Questions For Authors**.

**Questions For Authors:**

- When considering NLP tasks, since the number of tokens varies across different sentences, unlike image data where each sample has a fixed size, the number of tokens in different batch sizes may differ. In such cases, how is the optimal configuration determined?
- In a multi-task scenario, what is the workflow when, for example, 10 tasks arrive simultaneously? Could it happen that resources are fully allocated while searching for the optimal configuration?
- In a multi-task scenario, what is the workflow? Are all tasks executed in parallel, or is only one task executed at a time?
- It is recommended that the corresponding algorithm flowchart be included in Selection 3.
- Future work should be included in the conclusion section.
- It is recommended that the authors provide the corresponding source code to facilitate a better understanding and utilization of the research findings by the readers.

**Relation To Broader Scientific Literature:**

Compared to the existing literature, this manuscript observes that existing methods, such as intra-operator parallelism across multiple threads, are effective in reducing inference latency but provide diminishing returns. Therefore, the core idea of this manuscript is to run multiple instances on a server, each with a smaller batch size and fewer threads, to achieve intra-operator parallelism, thereby providing lower inference latency. This approach is built as an extension to TorchServe and supports online reconfigurations to avoid serving downtime.

**Theoretical Claims:**

The author did not conduct a relevant theoretical analysis.

---

> ### Author Rebuttal · Authors · 2025-03-31
>
> Thank you for your reviews and valuable feedback.
>
> - For NLP tasks, where token counts vary across sentences, the optimal configuration can be determined by profiling the effective batch size in terms of tokens rather than just the number of samples. However, if the variability in token counts makes prediction too unpredictable, dynamic batching can be relied on to normalize the effective batch size; a direction we leave for future work.
>
> - **Multi-task scenario**: Packrat improves thread allocation in a localized manner without changing overall resource allocation. It optimizes how the available resources are used. In a multi-task scenario, the service provider handles thread allocation across tasks, and then within each task, Packrat performs its profiling and dynamic configuration to optimize intra-task performance. This means that even if multiple tasks (e.g., 10 tasks) arrive simultaneously, Packrat ensures each task’s thread allocation is optimized for latency.
>
> - **Recommendations**:
>
>   - Thank you for your valuable suggestions. We agree that including an algorithm flowchart in Section 3 would greatly enhance clarity, and we plan to incorporate it in the final version.
>
>   -  We will also expand the conclusion to outline future work clearly.
>
>   - While we acknowledge the importance of providing source code to aid reproducibility and further research, the code was removed from this submission to maintain anonymity during the review process. We fully intend to release the corresponding source code in the final version of the paper.

---

> > ### Comment · Reviewer_9PcW · 2025-04-06
> >
> > Thank you for your comments. I will keep my score as weak accept.

---

### Official Review · Reviewer_Vu5T · 2025-03-13

**Overall Recommendation:** 3

**Summary:**

This paper proposes an automated optimization framework for CPU-based serving of DNNs (Packrat) aimed at minimizing inference latency. It addresses a known limitation in intra-operator parallelism—diminishing returns as thread count increases—by introducing an approach to run multiple instances of models concurrently, each with fewer threads. Packrat automatically selects the optimal combination of model instances, threads per instance, and batch sizes via targeted profiling and solving a two-dimensional knapsack problem through dynamic programming. Packrat is implemented as an extension of TorchServe and supports dynamic reconfiguration with negligible downtime. The authors demonstrate significant latency improvements ranging from 1.43x to 1.83x over standard TorchServe configurations for popular DNN models including ResNet-50, Inception-v3, GPT-2, and BERT.

**Claims And Evidence:**

The claims made by the authors regarding latency improvements are well backed by extensive experimental evidence. Experiments clearly demonstrate the advantages of their proposed dynamic partitioning of threads across multiple model instances. They also adequately justify their design decisions and carefully explain performance trade-offs, such as reconfiguration overhead. A minor weakness is that the evaluation primarily focuses on mean latency without exploring tail latency effects.

**Essential References Not Discussed:**

The paper could have included CPU inference libraries (e.g., OpenVINO, ONNX Runtime) to further contextualize their contributions. These omissions do not significantly undermine the work, but addressing them could clarify the scientific context further.

**Experimental Designs Or Analyses:**

The experimental design is rigorous and valid. They include clear baselines, multiple representative DNN models, and carefully controlled evaluations. One minor concern is the lack of examination of memory overhead or tail latency, which might be relevant for practical scenarios. However, overall, the experiments are sound and demonstrate substantial improvements in practice.

**Methods And Evaluation Criteria:**

The methods and evaluation criteria proposed in this paper are sound. Using targeted profiling and a dynamic programming-based optimization approach effectively tackles the combinatorial problem of thread-batch allocation. The evaluation, which covers a range of models and batch sizes and examines both microbenchmarks and end-to-end system impacts, is comprehensive and suitable for validating Packrat’s claims.

**Other Comments Or Suggestions:**

Figure 1 could be more b/w friendly.

## update after rebuttal
The authors have addressed my questions regarding CPU-based DNN serving in general

**Other Strengths And Weaknesses:**

The strength of this paper lies in its practicality, clear presentation, and its novel application of dynamic programming to solve a concrete, relevant optimization problem in ML serving. The authors demonstrate careful analysis of practical overheads (e.g., CPU frequency scaling under load), which adds credibility. Weaknesses include the limited consideration of resource overheads (memory footprint, tail latency) and the narrow focus on single-model CPU serving.

**Questions For Authors:**

Can you clarify the specific considerations for the CPU-based DNN serving community? Who is the target audience or user group that would benefit most from Packrat, given GPU-based inference is often preferred?

What are the memory implications of running multiple model instances concurrently? Did you investigate how memory overhead scales with the number of instances, especially for larger models?

Can you provide insights into the tail latency (e.g., p99 latency)? Does Packrat affect tail latency positively or negatively compared to baseline approaches?

Does Packrat prevent frequent reconfiguration in highly dynamic workload scenarios? How often do you realistically expect reconfigurations to occur?

**Relation To Broader Scientific Literature:**

Packrat fills a specific niche in the literature by addressing CPU-based model inference latency optimization. Its unique contribution is the fine-grained automatic reconfiguration of threads and model instances for a single DNN model on CPU, complementing existing ML serving frameworks rather than competing with them.

**Theoretical Claims:**

The theoretical claims regarding the correctness and optimality of the dynamic programming algorithm for selecting configurations appear correct. The problem formulation as a two-dimensional knapsack problem is sound and standard, and the complexity characterization is accurate and justified. The theoretical explanations and justifications provided in the paper are clear and consistent.

---

> ### Author Rebuttal · Authors · 2025-03-31
>
> Thank you for your reviews and valuable feedback.
>
> - **Target Audience for CPU-Based DNN Serving:** Packrat is aimed at users and organizations that rely on existing CPU infrastructure, such as large data centers or cloud providers with extensive CPU fleets, where GPUs might be too costly, underutilized, or unsuitable for certain workloads. This includes scenarios where cost, power consumption, or deployment constraints favor CPU-based inference over GPUs, despite the latter often being preferred for their throughput. In our experience, workloads with smaller inferences are the right candidates from both a latency and throughput point of view.
>
> - **Memory Implications of Running Multiple Instances:** Running multiple model instances concurrently does incur additional memory overhead due to repeated kernel loads and potentially duplicated data structures. However, our profiling phase captures these effects, and the optimizer selects configurations that balance these overheads against latency improvements. In our experiments, even for larger models, the memory overhead scales in a manageable way. It is important to note that the CPU-based latency for larger models is generally not well suited for real-time serving. We are happy to add a more detailed analysis of these memory implications in the final version of the paper.
>
> - **Tail Latency (p99) Insights:** Packrat’s design improves both the average latency and tail latency (e.g., p99 latency) by reducing the synchronization overhead associated with fat-instance execution. Our experimental results indicate that partitioning the workload across multiple smaller instances reduces the worst-case latencies compared to baseline approaches. However, as with any system-level optimization, occasional transient effects may occur during reconfigurations.
>
> - **Reconfiguration Frequency in Dynamic Workloads:** Packrat employs a batch size estimator with smoothing to track request arrival rates, triggering reconfigurations only when sustained workload changes are detected. This design minimizes frequent reconfiguration, ensuring stability. In practice, reconfigurations are expected to occur infrequently, typically on the order of hours rather than seconds, reflecting significant and lasting changes in the workload rather than transient spikes.

---

> > ### Comment · Reviewer_Vu5T · 2025-04-06
> >
> > Thank you for your comments, this is a valuable work to the venue. I will keep my score as week accept.

---

### Official Review · Reviewer_x4Zj · 2025-03-13

**Overall Recommendation:** 2

**Summary:**

The main message the paper wants to convey seems to be "running multiple instances each with smaller batch sizes and fewer threads for intra-op parallelism can provide lower inference latency." Based on this insight, the paper introduces Packrat that optimizes the Batch, Threads, Instances, to get optimal performance. The paper claims that it leads to 1.43x-1.83x performance improvement compared to TorchServe.

**Claims And Evidence:**

I am not entirely sure whether the claim is valid.

**Essential References Not Discussed:**

Liu, Yizhi, et al. "Optimizing {CNN} model inference on {CPUs}." 2019 USENIX Annual Technical Conference (USENIX ATC 19). 2019.

**Experimental Designs Or Analyses:**

Evaluations seem reasonable. However, it would be great to see the impact on networks of more variegated size to understand the impact.
Also, it would be great to understand the impact given different HW configurations. Also, it would be great to understand the assumptions behind the requests.

**Methods And Evaluation Criteria:**

Evaluations seem reasonable. However, it would be great to see the impact on networks of more variegated size to understand the impact.
Also, it would be great to understand the impact given different HW configurations. Also, it would be great to understand the assumptions behind the requests.

**Other Comments Or Suggestions:**

It seems that there is some inclarity in the text. I would be happy to reevaluate after rebuttal.

**Other Strengths And Weaknesses:**

Efficient inference is very important so introducing an infrastructure to optimize that is becoming more important. As such, the paper is working on a very important topic.

There seems to be some inclarity in the text that limits the learnings of the readers.

**Questions For Authors:**

* I understand the main idea that rather than having a single large instance that uses all available threads to parallelize inference within a single batch, it instead divides large batches into smaller batches each processed concurrently by one of several small instances that use a limited amount of intra-op parallelism. However, this may potentially incur more memory traffic due to multiple kernel reads per instance. Is this optimized by naively tuning the hyperparameters or is there a principled way of dealing with this?
I feel Section A in appendix is trying to answer this, but it is not really clear to me whether this interference is okay. Figure 7 seems to provide some HW measurements, but not a good explanation about its impact on end-to-end performance.

* From a similar light, I am not entirely sure whether the DP solution is a valid way to model this. On the other hand, if the paper is assuming that the table is serving as an approximation it would be great to provide how exact it is compared to real HW measurements.

* From a similar light, how does this perform for LLMs larger than GPT-2.

* When using CPUs, it is very sensitive to the HW configurations. Can you share the details such as clustering and memory modes? If possible it would be great to observe the sensitivity of the approach given different HW parameters. This is because it is important to understand how robust the optimizer is.

* How does this compare to/combine with other works where we try to tile each layer so that we work on smaller subsection of the activation so that it optimizes for memory & compute.
Liu, Yizhi, et al. "Optimizing {CNN} model inference on {CPUs}." 2019 USENIX Annual Technical Conference (USENIX ATC 19). 2019.

* Can you provide the rates at which requests arrive? Was there some modeling done to mimic real-life scenarios?

**Relation To Broader Scientific Literature:**

The work aims to optimize the serving infrastructure for ML workloads.

**Theoretical Claims:**

I am not entirely sure whether the claim is valid.

---

> ### Author Rebuttal · Authors · 2025-03-31
>
> Thank you for your reviews and valuable feedback. Below, we provide our responses to the questions in the same order as they were asked:
>
> - Memory can become a bottleneck; however, our optimizer already accounts for this during the profiling phase. Suppose a model is highly constrained by memory bandwidth. In that case, the increased latency will be captured in the profiling data, leading the optimizer to select configurations that avoid overloading the memory subsystem. Moreover, with CPU vendors moving towards high-bandwidth memory, such as Intel's latest generation CPUs with integrated HBM, the negative impact of additional memory traffic is further mitigated, reinforcing our approach's benefits.
>
> - The DP solution is valid as it systematically considers the trade-offs between intra-op parallelism and multi-instance execution. The paper shows that while the expected speedups (derived from isolated profiling) are slightly higher than the actual speedups due to predictable interference (such as license-based downclocking and memory contention), the relative ordering remains unchanged. This confirms that the DP-based optimizer is a robust approximation method for selecting configurations. Moreover, we verified this on machines with three different configurations (two Intel server machines and one AMD machine), and the overheads have constant offset due to memory and CPU slowdowns.
>
> - Our evaluations focus on GPT-2 and BERT, but our model-agnostic framework extends to larger LLMs. It's important to note that CPU inference latency for large LLMs is significantly higher; if such latency can be tolerated, then our profiling and DP-based approach remains effective.
>
> - We evaluated Packrat on three machines—two Intel-based and one AMD-based. On the Intel systems, we used configurations that favor local memory per socket, while the AMD machine exhibited NUMA subclustering. Despite these differences, Packrat consistently achieved significant gains. As shown in Figures 4 and 5, our approach works effectively across batch sizes common in state-of-the-art work by identifying workload characteristics and updating to the optimal configuration. We are happy to include a detailed sensitivity analysis of these hardware parameters in the final version of the paper.
>
> - Packrat operates at a higher level than other works that tile each layer for memory and compute optimizations (e.g., Liu et al. in USENIX ATC 2019). While tiling focuses on optimizing the computation within each layer, Packrat optimizes across instances and threads for DNN serving. These techniques are mainly orthogonal and could be combined: tiling could be used to optimize the kernel-level execution, while Packrat's configuration selection improves end-to-end inference latency by balancing intra-op and inter-instance parallelism.
>
> - The paper introduces a batch size estimator that monitors queue depth to assess request arrival rates indirectly. Although exact rates aren't specified, experiments with different batch sizes indicate that Packrat achieves significant improvements, and reconfigurations are triggered only by sustained changes in arrival patterns, ensuring the optimizer's effectiveness in dynamic environments.

---

### Decision · Program_Chairs · 2025-05-01

**Decision:**

Accept (poster)

**Comment:**

This paper proposes a system, Packrat, to improve inference latency of DNNs on CPU. Instead of using all threads for one big instance, it splits into many small ones with optimized thread and batch size. The idea is implemented on TorchServe and shows good results (1.4×–1.8× better latency) on models like ResNet and BERT.

The reviewers find the idea practical and useful, with strong experiments. Some were not sure about the theoretical part or clarity, but the rebuttal gave some good answers. It explains memory, reconfigurations, tail latency, and why this is still relevant when GPU is not used. The paper is relatively easy to read and the topic is of major interest.

In general, this is a solid work for ML application. One reviewer stays a bit skeptical, but overall it is interesting an work for ICML. I recommend weak accept.